# Translation, adaptation and psychometric evaluation of the German version of the Abortion Attitude Scale – A secondary analysis of a cross-sectional study among medical students

Anja Lindig[1,2]*, Eva Christalle[2], Mareike Rutenkröger[2], Jördis Maria Zill[2], Mirja Baumgart[2]

1 Department of Psychology, Carl von Ossietzky University Oldenburg, University Medicine Oldenburg, Oldenburg (Oldb), Lower Saxony, Germany, 2 Department of Medical Psychology, University Medical Center Hamburg-Eppendorf, Hamburg, Germany

* anja.lindig@uni-oldenburg.de

## Abstract

### Background

Unintentionally pregnant individuals in Germany seeking an abortion face challenges due to legal regulations, stigma and difficult access to abortion care. Abortion attitudes of (prospective) physicians influence the care situation. To measure these attitudes, psychometrically sound instruments like the Abortion Attitude Scale (AAS) are necessary. So far, no instruments assessing attitudes toward abortions are available in German. This study aims to translate, culturally adapt and psychometrically test the AAS.

### Methods

This is a secondary analysis of a cross-sectional study on abortion attitudes of medical students in Germany. The English 14-item AAS was translated into German and adapted using a team translation protocol. Comprehensibility was tested via cognitive interviews ($n = 10$ medical students). We analyzed acceptance (completion rate), factor structure (confirmatory factor analysis (CFA), model fit), item characteristics (response distribution, item difficulties, corrected item-total correlations, inter-item correlations), and reliability (McDonald's omega).

### Results

The translated and adapted AAS version was comprehensible. AAS data of 305 medical students could be included in analysis. Completion rate was above 98% for all items. The CFA results confirmed a one-factor structure, but a model without item 10 and correlations between item 8 and item 13 showed the best model fits. Floor or

**Data availability statement:** The dataset collected and analyzed during this study is available on the OSF data repository (https://doi.org/10.17605/OSF.IO/MQ3SN).

**Funding:** This study was planned and conducted as part of the study "Person-centeredness in healthcare and support services for women with unwanted pregnancy" (CarePreg), which was funded by the German Federal Ministry of Health (Bundesministerium für Gesundheit, BMG) with the grant number: 2520FSB113. Dr. Jördis Zill (co-author) was principal investigator of the CarePreg Study and received the funding. The funders had no role in study design, data collection and analysis, decision to publish, or preparation of the manuscript.

**Competing interests:** The authors have declared that no competing interests exist.

**Abbreviations:** AAS, Abortion Attitude Scale; AIC, Akaike Information Criterion; AL, Anja Lindig; AMOS, Analysis of Moment Structure, Statistical Package for the Social Sciences, International Business Machines Corporation; BIC, Bayesian Information Criterion; CFA, Confirmatory Factor Analysis; Chi², Discrepancy Chi-squared Statistic; CFI, Comparative Fit Index; COSMIN, Consensus-based Standards for the Selection of Health Measurement Instruments; Df, Degree of freedom; EC, Eva Christalle; JZ, Jördis Zill; KMO, Kaiser-Meyer-Olkin criterion; MB, Mirja Baumgart; MLR, maximum likelihood estimator; MR, Mareike Rutenkröger; RMSEA, Root Mean Square Error of Approximation; SPSS, Statistical Package for the Social Sciences, International Business Machines Corporation; SRMR, Standardized Root Mean Squared Residual; TLI, Tucker-Lewis Index; TRAPD, Translation, Review, Adjudication, Pretesting, Documentation

ceiling effects could be found for 7 items, item difficulties ranged between 0.39 and 0.94, corrected item-total correlations ranged between 0.460 and 0.766 for the best model, inter-item correlations ranged between.129 to.681, and McDonald's omega was above 0.9 for both models.

## Conclusion

The German AAS is a brief measure that was accepted and demonstrated adequate psychometric properties. Suboptimal performance of some items indicated the need for further validation, particularly of the 13-item version. The AAS can improve evaluation of abortion attitudes in Germany. This may support identification of barriers in healthcare provision and improve care for unintentionally pregnant individuals.

## Introduction

Attitudes towards abortion are a key factor in understanding societal trends in opinions, the impact of gender-based violence, and legislative changes surrounding reproductive rights in various countries [1]. According to the Theory of the Triangle of Violence [2], negative attitudes towards abortion can manifest as gender-based violence across three interconnected dimensions: 1) *structurally*, restrictive abortion legislation in some regions violate human rights [3] 2) *culturally*, stigmatizing attitudes often stem from traditional and conservative values, such as prioritizing responsibility and caregiving over self-determination or reinforcing archetypes of femininity that equate womanhood with motherhood, 3) through dire*ct violence*, expressed via individual actions and discriminatory treatment. These dimensions collectively contribute to abortion stigma, which extends beyond social dynamics by significantly impacting women's mental health. Women experiencing stigmatization report higher levels of depression, anxiety, stress, psychological distress, and somatic symptoms [4–6]. The community's attitudes towards abortion, therefore, directly influence women's health and internalized stigma, emphasizing the need for systematic exploration of these attitudes [1].

Conscientious objection to abortion adds another dimension to this topic. Defined as a physician's refusal to perform legal abortions based on religious or moral beliefs, it is the only avenue for providers to refuse healthcare that would typically fall within their scope of practice [7]. Conscientious objection is criticized for inadequately balancing abortion access with providers' rights [7]. It has been identified as a barrier to timely and safe abortion care, contributing to increased morbidity and mortality [8,9]. Studies indicate that, depending on the country and its legal regulations, 14–80% of clinicians globally refuse to provide legal abortion services [8]. Conscientious objection exists also in Germany (based on §12 SchKG, [10]) and results in stigma against people seeking abortions and healthcare professionals working in abortion care [11–16] as well as limited access to abortion care [16,17]. German law allows abortion under certain conditions while maintaining its status as technically illegal under the German criminal code (§218 StGB, [18]). These include: 1) within the first

14 weeks after conception and after undergoing mandatory counselling and a 3-day waiting period between counselling and abortion, 2) for medical reasons in case of expected psychological or physiological harm to the mother, and 3) for criminal reasons like sexual abuse. This regulatory framework created a nuanced and often polarized societal discourse [19–21]. The lack of training on abortion procedures in most German medical school curricula reinforces the barriers for high-quality abortion care in Germany [15,16]. Thus, physicians often bear the responsibility of acquiring the necessary skills for abortion care independently. While the total number of abortions performed in Germany has remained stable (106.218 reported in 2023 [22])), the number of facilities offering abortion services ("Meldestellen") has drastically declined over the past 20 years, dropping from 2.050 to 1.092 [23]. A recent qualitative study involving 18 experts in psycho-social and medical abortion care highlighted that this shortage of providers significantly hampers the delivery of person-centered abortion care [16].

In this context, the need for psychometrically reliable tools to evaluate attitudes towards abortions becomes evident. Such tools are essential for addressing the complex interplay of stigma, conscientious objection, and reproductive rights to inform interventions or policies for future high-quality abortion care, particularly within specific sociocultural contexts like Germany. There is especially a high demand for evaluating attitudes of (future) physicians as a predictor for their (future) participation in abortion care [24]. The theory of planned behavior postulates that attitudes (towards abortion), social norms and perceived control of action influence intentions (to perform abortions) and that perceived control of action predicts behavior [25]. Accordingly, physicians having more positive attitudes towards abortion are more likely to provide abortions [26,27].

Existing self-report instruments on abortion attitudes, such as the Stigmatizing Attitudes, Beliefs, and Actions Scale [28] or the Abortion as a Right Scale [29] have been developed primarily in African countries like Ghana and Zambia. Still, the item content and focus of the scales do not align with the sociocultural realities of countries like Germany. Others, like the Abortion-Providing Physicians Scale [30] or the Adolescent Attitudes to Abortion Scale [31] have a different focus by assessing attitudes of health professionals towards abortion providers or of adolescents. Furthermore, many instruments evaluate the level of agreement with abortion in a series of circumstances and not cognitions and beliefs [1,32]. The scarcity of instruments with robust psychometric properties highlights a gap in tools designed for specific populations, such as those in Germany [1]. Since there is no instrument available on abortion attitudes in the German language, translating the Abortion Attitude Scale (AAS) into German is promising. The original has demonstrated good reliability and construct validity, is accessible to the German sociocultural background, and targets cognitions and beliefs [33]. The AAS, developed by Linda A. Sloan in the United States in 1983, captures a range of perspectives on abortion, from personal beliefs to perception of societal norms and is usable in various settings [33,34]. Alspaugh et al. (2021) used the AAS to assess the attitudes of women's health and neonatal nurses in the U.S., applying only slight modifications to reflect a more neutral, person-centred language, without systematic adaptation or testing of the adapted items [35]. Furthermore, they did not assess psychometric properties of the scale.

It is necessary not only to translate but also adapt an instrument to ensure that it maintains its conceptual and measurement equivalence in the target culture [36], since existing measures may not fully capture the cultural nuances and legal realities of other contexts. This includes potential differences in language semantics, cultural norms, and value systems that may influence how respondents interpret and respond to survey items [36]. Furthermore, there is an increasing need for comparative research on abortion attitudes across countries. For such studies to be meaningful, the required tools must be psychometrically robust across different settings and exhibit similar characteristics.

Therefore, this study aimed to translate and culturally adapt the AAS instrument into German and evaluate its psychometric properties.

## Methods

### Study design

We conducted this secondary analysis as part of a cross-sectional online study on factors influencing attitudes towards abortion of medical students in Germany [37]. Based on the Intergroup Contact Theory [38] and the Theory of Planned

Behavior [25], the analyzed influencing factors included subjective and objective knowledge about abortions, contact experiences with people seeking an abortion, fear of stigmatization (if participating in abortion care in the future), empathy, subjective norms, perceived behavioral control, and willingness to participate in abortion care. Abortion attitudes were measured using the AAS.

The cross-sectional online study, including the secondary analysis at hand, was preregistered on AsPredicted (registration number 142964, https://aspredicted.org/b5b2q.pdf).

## Measure

The 14-item AAS was initially developed for use by health educators helping students to explore their values concerning abortion. The author validated the scale by assessing interrater agreement and reliability estimates. Factor analysis provided a unidimensional structure [33]. When applying the scale to high school and college students, active "Right to Life" members, and abortion service associates, the AAS showed a high reliability ($\alpha = .92$) and high construct validity [33]. Furthermore, the instrument effectively distinguished between "Right to Life" members and abortion service associates [33].

Items of the AAS are rated on a Likert-scale from 0 ("strongly disagree") to 5 ("strongly agree") [33]. For the original English items of the 14-item scale, see Table 1. To calculate a sum score, the scales of items 1, 3, 4, 7, 9, 12, and 14 need to be reversed. Higher AAS scores indicate a stronger pro-abortion attitude. According to Sloan (1983), participants can be grouped to one of five categories according to their sum score: strong contra abortion (original wording: strong pro-life, sum score 0–15), moderate anti-abortion (original wording: moderate pro-life, sum score 16–26), unsure (sum score 27–43), moderate pro-abortion (sum score 44–55), strong pro-abortion (sum score 56–70).

## Translation

To translate and adapt the English AAS into German, we employed the team translation protocol TRAPD (Translation, Review, Adjudication, Pretesting, and Documentation). Initially, two authors (AL, MB), both fluent in German and English, independently translated the English AAS into German. They also proposed adaptations of items when necessary due to cultural or legal differences in the (political) context. Subsequently, a third team member (MR) reviewed the translations, choosing one version or suggesting a third. In a final discussion, AL, MB, and MR agreed on the translated and adapted German version [39–41].

## Assessment of comprehensibility

Following the COSMIN criteria (Consensus-based Standards for the Selection of Health Measurement Instruments) [42], we assessed comprehensibility by evaluating whether the AAS items were understood by medical students as intended. We conducted three rounds of cognitive interviews with a convenience sample of $n = 10$ medical students currently studying human medicine at a medical school in Germany. The interviews, led by MB, were conducted online via Zoom and audio-recorded. Participants' demographic characteristics were assessed, and descriptive statistics were calculated using SPSS (IBM SPSS Statistics, Version 29.0.1.0). An interview guide was created based on the recommendations of Willis et al. [43], incorporating verbal probing techniques and paraphrasing. The interview guide included questions such as "What do you understand by abortion?", "How do you understand this statement?", "What do you understand by murder in this item?". Alternatively, if different versions of an item were presented: "Is this version easier to understand than the first version?", "Does the question mean the same thing?". Following each round of cognitive interviews, AL and MB analyzed and discussed the feedback and suggestions from the participants. We revised the German AAS items accordingly and tested the revised versions in the next round.

We further send the final survey of the cross-sectional study (including the AAS) to our cooperation partners, who supported the study team during their research on abortion care. We received feedback from $n = 2$ abortion counselors and $n = 3$ gynecologists providing abortion care. According to their feedback, further modifications of the AAS were not necessary.

**Table 1. Comparison of the original AAS scale and the German AAS scale provided in a translated version.**

|  | Original scale | German AAS [translated]* |
|---|---|---|
| Introduction | This is not a test. There are no wrong or right answers to any of the statements, so just answer as honestly as you can. The statements ask you to tell how you feel about legal abortion (the voluntary removal of a human fetus from the mother during the first three months of pregnancy by a qualified medical person). Tell how you feel about each statement by circling one of the choices beside each sentence. Here is a practice statement: Abortions should be legalized. strongly agree – agree – slightly agree – slightly disagree – disagree – strongly disagree Please respond to each statement and circle only one response. No one else will see your responses without permission. | This is not a test. There are no wrong or right answers to any of the statements, so just answer as honestly as you can. The statements are about your views on **abortion during the first three months of pregnancy by a qualified health professional. Please indicate how much you agree or disagree with a statement by checking an answer option.** Here is an example: Abortions should be legalized. strongly agree – agree – slightly agree – slightly disagree – disagree – strongly disagree Please respond to each statement and **select only one response** at a time. No one else will see your responses without permission. |
| Item 1 | The supreme court should strike down legal abortions in the United States. | **In Germany, abortions should be banned under all circumstances.** |
| Item 2 | Abortion is a good way of solving an unwanted pregnancy. | Abortion is a good way of solving an unwanted pregnancy. |
| Item 3 | A mother should feel obligated to bear a child she has conceived. | **A person** should feel obliged to bear a child **that was conceived**. |
| Item 4 | Abortion is wrong no matter what the circumstances are. | **Abortions** are wrong no matter what the circumstances are. |
| Item 5 | A fetus is not a person until it can live outside its mother's body. | A fetus is not a person until it can live outside **the pregnant person's** body. |
| Item 6 | The decision to have an abortion should be the pregnant mother's. | The decision **for an abortion** should be the **pregnant person's**. |
| Item 7 | Every conceived child has the right to be born. | Every conceived child has the right to be born. |
| Item 8 | A pregnant female not wanting to have a child should be encouraged to have an abortion. | A **pregnant person** not wanting to have a child should be encouraged to have an abortion. |
| Item 9 | Abortion should be considered killing a person. | Abortion should be considered killing a person |
| Item 10 | People should not look down on those who choose to have abortions. | People should not look down on those who choose to have **an abortion**. |
| Item 11 | Abortion should be an available alternative for unmarried, pregnant teenagers. | Abortion should be an **easily accessible alternative for pregnant minors**. |
| Item 12 | Persons should not have the power over the life or death of a fetus. | **No one** should have the power over the life or death of a fetus. |
| Item 13 | Unwanted children should not be brought into the world. | Unwanted children should not be brought into the world. |
| Item 14 | A fetus should be considered a person at the moment of conception. | A fetus should be considered a person at the moment of conception. |

Note: Changes of the German scale compared to the English scale are marked in bold letters. The German AAS scale was translated into English by the first author alone, without a team translation approach, solely for the purpose of readability within this manuscript. The German AAS translated into English is not a systematic back translation, not validated and should therefore not be used without further testing.

## Data collection

Data for the cross-sectional study were collected using Unipark (EFS Survey, Tivian XI GmbH) between September 8th and October 4th 2023. We included medical students who were at least 18 years old and enrolled at a medical school in Germany at the time of participation. Medical student councils from all medical faculties in Germany were contacted via

email and asked to resend the study invitation and a link to the survey to their students or through online postings on the faculty bulletin boards. Additionally, local groups of Medical Students for Choice (MSFC), an international initiative to support safe abortions, distributed the survey link via internal faculty email lists and Instagram channels. Participation was voluntary and without financial incentives. Participants were informed about the study's purpose and background, as well as the procedure for data storage and analysis. They provided informed consent by ticking the respective box after reading the study information. The AAS was the first instrument participants completed within the survey, followed by several other items and instruments [37]. Demographic characteristics were assessed at the end of the survey.

## Psychometric data analyses

We excluded cases if: 1) they were entirely empty, 2) had more than 30% missing AAS items [43], 3) participants did not meet the inclusion criteria, and/or 4) answered one of the two control questions incorrectly. These control questions were used to identify careless responses [44,45]. We further screened cases for suspicious response behaviour (e.g., participant always chooses the same response option) and excluded suspicious cases after discussions within the study team [44–46].

For the included participants, we calculated descriptive statistics of their demographic characteristics. For acceptance analysis, we examined the completion rate (percentage of missing values) per item.

In the next step, we conducted item analyses. This involved examining response distribution, item difficulty (calculated by standardizing the mean to range from 0 to 1 by division by the maximum possible value of 5, recommended range 0.2–0.8) to assess floor and ceiling effects (item difficulties below 0.2 indicate a floor effect; item difficulties above 0.8 indicate a ceiling effect) [47]. Additionally, we calculated item means and standard deviations, corrected item-total correlations (recommended range >0.3), inter-item correlations (recommended range >0.3), and sum scores [46–48]. For analysing corrected item-total correlations and inter-item correlations, missing data were replaced by multiple imputation [48–51].

To assess structural validity, we conducted a confirmatory factor analysis (CFA), which is the preferred approach when a hypothesis about the dimensionality is available [52]. First, we examined the suitability of factor analysis, by analyzing the Kaiser-Meyer-Olkin (KMO) measure of sampling adequacy and Bartlett's test of sphericity [53,54]. KMO value should be higher than.50 and Bartlett`s test value should be less than.05 to fulfil the criteria for calculating a factor analysis [53,54]. The author of the original AAS postulated a one-factor structure of the AAS [33]. However, the original publication lacks information on factor loadings or fit indices for this assumption.

Furthermore, other studies using the AAS did not analyze the factor structure of the instrument but relied on the assumption of a one-factor-structure [34,35]. Thus, a previous hypothesis about the dimension of the construct is available and a CFA is therefore appropriate [52]. We initially hypothesized a one-factor structure for the German AAS and modified the model step-by-step based on the results of the factor analysis, item analysis and content of specific items. We conducted a CFA with a unidimensional model (model 1) [55–57] using a robust version of maximum likelihood estimator (MLR) and full information maximum likelihood to address missing values. There are various criteria for interpreting factor loadings and determining cut-offs for appropriate factor models [58,59]. For this study, we chose to use the established, less conservative criterion of.40 as the cut-off for acceptable factor loadings in samples larger than $n = 200$ [58]. To evaluate model fit of each model, a range of global goodness-of-fit indices were calculated. We used established criteria to interpret the fit of the estimated models [60,61]: discrepancy chi-squared statistic (Chi²)/degree of freedom (df) (ratio between 2 and 3 indicates good fit), Comparative Fit Index (CFI; values ≥0.90 indicate good fit), Tucker-Lewis Index (TLI; values ≥0.90 indicate good fit), Root Mean Square Error of Approximation (RMSEA; recommended value ≤0.06), and Standardized Root Mean Squared Residual (SRMR; recommended value <0.08). For comparison of models, we used Akaike information criterion (AIC) and Bayesian Information Criterion (BIC), for both of them a smaller value indicates a better fit [60,61]. Afterwards, we assessed reliability by McDonald's omega, here values above 0.70 indicate acceptable internal consistency [62].

Analysis of demographic data, completion rate, and item analysis were done using SPSS (IBM SPSS Statistics, Version 29.0.1.0). We used R Version 4.3.2 (R Core Team, Vienna, Austria) for CFA (specifically, lavaan package [63]) and reliability analyses (specifically, semtools [64]).

### Ethics approval and consent to participate

This study was conducted in accordance with the latest version of the Helsinki Declaration of the World Medical Association, principles of good scientific practice, and research ethics standards. It was approved by the Local Ethics Committee of the University Medical Centre Hamburg-Eppendorf, Germany (LPEK-0628, May 2023). Data protection and confidentiality requirements were adhered to. Participation was voluntary. All participants received information about the aims of the study, data collection, and how the data would be used, and gave informed consent before participation.

## Results

### Translation and adaptation

Both translators (AL and MB) and the reviewer (MR) had similar translations of the ASS. S1 Appendix (Table A) provides an overview of the original AAS, the results after translation and adaptation, and the final German instrument. During the first round of team discussion, we reached consensus on the translation and adaptation of all items, the response scale, and the introduction. In the introduction, we decided to remove the term "legal" from "legal abortions" because abortions are illegal under the German criminal code (though they will not be punished under certain conditions, §218 StGB [65]). Item 1 was adapted to align with German policy and national legal restrictions. Here, the term "legal abortions" might also be misleading in the German context, so we decided on the term "under all circumstances" (dt. "unter allen Umständen"). For item 5, we added an alternative version that avoids negation to increase the item's comprehensibility. For the adaptation of item 6, we added an alternative version that slightly differs in word order. We discussed item 9 regarding the meaning of the original term "killing a person". In German, the two terms "killing" and "murder" ("Tötung/Totschlag" and "Mord") have different definitions according to the German criminal code (§211 and §212 StGB [66]). Those are comparable to the American legal system; their definitions are not laid down in a uniform set of rules but in respective criminal statutes of each state. Thus, we decided to discuss both versions, "killing" and "murder", with our participants. We also tested two versions of item 10 containing the singular and plural forms of the term "abortion". Item 11 had also been adapted culturally since the public view on the relevance of marital status of "unmarried pregnant teenager" has changed within the past 40 years. It might not be comparable between Germany and the United States. Additionally, abortion is actually an option for pregnant teenagers under German law. We decided to test two versions, which differ slightly in wording and the use of the term "unmarried". We adapted item 12 by changing the term "persons" (dt. "Personen") to "nobody" (dt. "niemand"), to avoid negation and improve readability. Finally, to make the instrument more inclusive, we changed the term "mother" or "female" to "(pregnant) person" in items 3, 5, 6, and 8. Pregnant person is also meant to include trans men and other individuals who would not call themselves a mother or female person, but can become pregnant. Finally, we decided to change two results categories from "moderate/strong pro-life" to "moderate/strong contra abortion" to avoid reproducing the nomenclature used by anti-abortion activists.

### Assessment of comprehensibility

We conducted cognitive interviews with $n = 10$ medical students. For demographic data of participants, see Table B in S1 Appendix.

The first round of cognitive interviews ($n = 3$) revealed that the adapted introduction, the response scale and items 2, 4, 7, 12, and 14 were well understood. However, those items were tested again in round 2 ($n = 2$) and 3 ($n = 5$) of cognitive interviews, but continued to be well understood by all participants and accordingly did not require further

adaptation. We decided to test another version for item 1 in rounds 2 and 3 (replacing "under all circumstances" (dt. "unter allen Umständen") with the term "basically" (dt. "grundsätzlich"). Participants expressed that "basically" is the "weaker" term than "under all circumstances" by not including all possible circumstances. To remain as close as possible to the original item, the study team decided to use "under all circumstances". After interview round 1, another version of item 3 using a passive term was tested ("she has conceived"/ dt. "das sie gezeugt hat" vs. "who was conceived"/ dt. "das gezeugt wurde"). Participants of the following two rounds preferred the second version. Almost all participants preferred the second version of the adapted items 5 and 6 because they were more comprehensible by avoiding negation and complex sentence structures. After the first round of cognitive interviews and based on suggestions by participants, we developed a new version for item 8, including the term "should be supported in having an abortion" (dt. "sollte bei einem Schwangerschaftsabbruch unterstützt werden") compared to "should be encouraged to have an abortion" (dt. "sollte zu einem Schwangerschaftsabbruch ermutigt werden"). Participants of rounds 2 and 3 discussed that this version has a more "active" and "directive" meaning. Thus, the study team decided for the first translated and adapted version to stick to the original item. Almost all participants preferred version 1 of item 9 because the term "killing" (dt. "Tötung/Totschlag") was described as less harmful compared to "murder" (dt. "Mord") and might therefore be less stigmatizing. Since the translation of the term "killing" is also closer to the original item, the study team decided to use the first version of the adapted item 9 with slight modifications in wording. For item 10, participants preferred the adapted item version 2, which uses the singular form of "abortion". Thus, the study team decided to use this version, even though it is less congruent with the original item. Based on discussions about item 11 during the first round of cognitive interviews, the study team decided to adapt the item again and test a new version, which combines elements of previously tested versions 1 and 2. This third version was well understood by all participants of interview rounds 2 and 3 and thus used for the final instrument. After the first round of cognitive interviews and suggestions by participants, we developed a new version of item 13, which included the term "should not be given birth to" (dt. "sollten nicht auf die Welt gebracht werden") instead of "should not be brought into the world" (dt. „sollten nicht in die Welt gesetzt werden"), since the new version is less offensive. This latest version was well understood by participants of rounds 2 and 3 and thus used for the final instrument. The final German AAS instrument can be found in S1 Appendix (Table A), a translated version is displayed in Table 1.

### Psychometric evaluation

**Data cleaning.** Before analysis, 46 cases were excluded because they were empty ($n = 35$), the inclusion criteria were not fulfilled ($n = 6$), the second control question was incorrectly answered ($n = 3$), and identical and suspicious responses for most of the scales ($n = 2$). Data from 305 medical students were available for further analysis.

**Sample characteristics.** Table 2 gives an overview of participants' demographic characteristics. Mean age of the 305 included medical students was 23.7 years (SD 3.81), and on average, they were in the 6th semester (SD 3.08). Most were female (63.3%), living in a permanent unmarried partnership (37.7%), currently residing in North Rhine-Westphalia (44.9%), and their preferred future specialisation was gynecology and obstetrics (11.8%). Most have no religious beliefs (38.4%) and never visit a religious site (31.1%). Eleven participants (3.6%) had already had an abortion themselves. Further details on participants' demographic characteristics (own country of birth, parents' country of birth, state of origin, hometown size, mother tongue, pregnancies, children, and confession) can be found in S2 Appendix (Table A).

Most participants could be grouped into the category "Strong pro-abortion" (47.87%) or "Moderate pro-abortion" (36.39%), 12.3% were grouped into the category "Unsure", and a minority showed "Moderate contra abortion" (2.9%) or "Strong contra abortion" (0.6%) attitudes (see Table 3).

**Acceptance.** All items had high acceptance. The highest rate of missing values was 4 missing values (1.32%) for Item 12 (see Table 4). Missing values could be observed for 9 cases, with one missing value per case. Accordingly, 97.05% of the respondents filled out the AAS completely.

**Table 2. Demographic characteristics of participants (*n* = 305 medical students).**

| | | *n* | % |
|---|---|---|---|
| Age | **Mean** | 23.7 | |
| | **SD** | 3.81 | |
| | ***n*** | 234 | |
| Semester | **Mean** | 6.67 | |
| | **SD** | 3.08 | |
| | ***n*** | 234 | |
| Gender | Female | 193 | 12.5 |
| | Male | 38 | 63.3 |
| | Divers/ non-binary | 1 | 0.3 |
| | Preferred not to answer this question | 2 | 0.7 |
| | Missings | 71 | 23.3 |
| Relationship status | Single | 105 | 34.4 |
| | Permanent partnership (unmarried) | 115 | 37.7 |
| | Married | 12 | 3.9 |
| | Divorced | 2 | 0.7 |
| | Missings | 71 | 23.3 |
| Current state of living | Baden-Wuerttemberg | 16 | 5.2 |
| | Bavaria | 12 | 3.9 |
| | Berlin | 2 | 0.7 |
| | Hamburg | 6 | 2.0 |
| | Hesse | 1 | 0.3 |
| | Lower-Saxony | 16 | 5.2 |
| | North Rhine-Westphalia | 137 | 44.9 |
| | Saxony | 38 | 12.5 |
| | Schleswig-Holstein | 5 | 1.6 |
| | Thuringa | 1 | 0.3 |
| | Missings | 71 | 23.3 |
| Had an abortion themselves | 0 | 219 | 71.8 |
| | 1 | 11 | 3.6 |
| | Missings | 75 | 24.6 |
| Future preferred specialization | General medicine | 28 | 9.2 |
| | Anesthesiology | 28 | 9.2 |
| | Anatomy | 1 | 0.3 |
| | Ophthalmology | 5 | 1.6 |
| | Dermatology | 3 | 1.0 |
| | Gynecology and obstetrics | 36 | 11.8 |
| | Otolaryngology | 2 | 0.7 |
| | Hematology | 1 | 0.3 |
| | Internal medicine | 23 | 7.5 |
| | Cardiology | 7 | 2.3 |
| | Pediatrics and adolescent medicine | 34 | 11.1 |
| | Child and adolescent psychiatry and psychotherapy | 3 | 1.0 |
| | Oral and maxillofacial surgery | 5 | 1.6 |
| | Neurosurgery | 12 | 3.9 |
| | Neurology | 10 | 3.3 |
| | Public health care | 1 | 0.3 |

*(Continued)*

|  | | n | % |
|---|---|---|---|
|  | Oncology | 6 | 2.0 |
|  | Pathology | 2 | 0.7 |
|  | Pharmacology | 1 | 0.3 |
|  | Psychiatry and psychotherapy | 6 | 2.0 |
|  | Psychosomatic medicine and psychotherapy | 1 | 0.3 |
|  | Radiology | 8 | 2.6 |
|  | Forensic medicine | 6 | 2.0 |
|  | Urology | 3 | 1.0 |
|  | Missing | 73 | 23.9 |
| Influence of faith | I have no religious beliefs | 117 | 38.4 |
|  | Not at all | 36 | 11.8 |
|  | Low | 43 | 14.1 |
|  | Somewhat | 19 | 6.2 |
|  | Rather strong | 15 | 4.9 |
|  | Very strong | 4 | 1.3 |
|  | Missings | 71 | 23.3 |
| Visit of a religious sites | Never | 95 | 31.1 |
|  | Once a year | 61 | 20.0 |
|  | Several times a year | 59 | 19.3 |
|  | Several times a month | 7 | 2.3 |
|  | Once a week or more often | 11 | 3.6 |
|  | Missings | 72 | 23.6 |

Note: SD = standard deviation

**Table 3. Sum scores and categories for the German AAS.**

|  | Sum score | n | % |
|---|---|---|---|
| Strong pro-abortion | 70−56 | 147 | 47.87 |
| Moderate pro-abortion | 55−44 | 110 | 36.39 |
| Unsure | 43−27 | 37 | 12.13 |
| Moderate anti-abortion* | 26−16 | 9 | 2.9 |
| Strong anti-abortion* | 15−0 | 2 | 0.6 |
|  | Total | 305 | 100 |

Note: Wording of the categories in the original publication were "strong pro-life" and "moderate pro-life". They were changed in conjunction with the translation process to be more neutral.

**Analysis of AAS items.** Table 4 shows response distribution, means and standard deviations, item difficulties, acceptance (completion rates), and corrected item-total correlations for the 14 items. Means ranged between 1.99 (item 8) and 4.78 (item 1) on a scale from 0 to 5. Accordingly, item difficulties ranged from 0.39 (item 8) to 0.96 (item 1), with 7 items exceeding 0.8 and thus demonstrating ceiling effects. Corrected item-total correlations ranged from 0.295 (item 10) to 0.770 (item 9) for model 1 and 0.469 (item 8) to 0.766 (item 9) for model 2, and inter-item correlations from .129 (item 1/ item 10) to .681 (item 7/ item 12, see S2 Appendix, Table B).

**Factor analysis.** Assumptions for factor analysis were met [42,53,54]. KMO measure was .932 and Bartlett's test of sphericity yielded $\chi^2 = 2016.04$, $p < .001$ [53,54]. For model 1, we assumed a one-factor structure without correlations between

**Table 4. Means, standard deviation, skewness, item difficulty, acceptance and item discrimination of the German AAS (n = 305 medical students).**

| | Strongly Disagree N (%) | Disagree N (%) | Slightly Disagree N (%) | Slightly Agree N (%) | Agree N (%) | Strongly Agree N (%) | Mean (SD) | Item difficulty | Acceptance (Completion rate in %) | Corrected item-total correlation Model 1 | Corrected item-total correlation Model 2 |
|---|---|---|---|---|---|---|---|---|---|---|---|
| Item 1 | 269 (88.2%) | 19 (6.2%) | 7 (2.3%) | 4 (1.3%) | 4 (1.3%) | 1 (0.3%) | 4.78 (0.72%) | 0.96 | 99.67 | 0.552 | 0.558 |
| Item 2 | 7 (2.3%) | 6 (2.0%) | 14 (4.6%) | 40 (13.1%) | 75 (24.6%) | 163 (53.4%) | 4.16 (1.17%) | 0.83 | 100.0 | 0.719 | 0.721 |
| Item 3 | 188 (61.6%) | 47 (15.4%) | 33 (10.8%) | 21 (6.9%) | 10 (3.3%) | 6 (2.0%) | 4.19 (1.26%) | 0.84 | 100.0 | 0.697 | 0.696 |
| Item 4 | 251 (82.3%) | 35 (11.5%) | 10 (3.3%) | 4 (1.3%) | 3 (1.0%) | 2 (9.7%) | 4.71 (0.78%) | 0.94 | 100.0 | 0.523 | 0.525 |
| Item 5 | 31 (10.2%) | 58 (19.0%) | 77 (25.2%) | 59 (19.3%) | 56 (18.4%) | 24 (7.9%) | 2.40 (1.44%) | 0.48 | 100.0 | 0.594 | 0.592 |
| Item 6 | 1 (0.3%) | 3 (1.0%) | 6 (2.0%) | 33 (10.8%) | 78 (25.6%) | 184 (60.3%) | 4.41 (0.87%) | 0.88 | 100.0 | 0.644 | 0.637 |
| Item 7 | 79 (25.9%) | 80 (26.2%) | 76 (24.9%) | 46 (15.1%) | 10 (3.3%) | 13 (4.3%) | 3.44 (1.34%) | 0.68 | 99.67 | 0.750 | 0.754 |
| Item 8 | 41 (13.4%) | 54 (17.7%) | 112 (36.7%) | 67 (22.0%) | 23 (7.5%) | 7 (2.3%) | 1.99 (1.21%) | 0.39 | 99.67 | 0.462 | 0.460 |
| Item 9 | 207 (67.9%) | 48 (15.7%) | 22 (7.2%) | 17 (5.6%) | 5 (1.6%) | 6 (2.0%) | 4.37 (1.14%) | 0.87 | 100.0 | 0.770 | 0.766 |
| Item 10 | 10 (3.3%) | 0 (0.0%) | 3 (1.0%) | 14 (4.6%) | 38 (12.5%) | 240 (78.7%) | 4.59 (1.03%) | 0.92 | 100.0 | 0.295 | – |
| Item 11 | 11 (3.6%) | 22 (7.2%) | 44 (14.4%) | 68 (22.3%) | 65 (21.3%) | 95 (31.1%) | 3.44 (1.43%) | 0.69 | 100.0 | 0.640 | 0.642 |
| Item 12 | 119 (39.0%) | 97 (31.8%) | 63 (20.7%) | 15 (4.9%) | 3 (1.0%) | 4 (1.3%) | 4.00 (1.05%) | 0.80 | 98.68 | 0.706 | 0.711 |
| Item 13 | 34 (11.1%) | 58 (19.0%) | 70 (23.0%) | 85 (27.9%) | 36 (11.8%) | 20 (6.6%) | 2.30 (1.38%) | 0.46 | 99.33 | 0.558 | 0.562 |
| Item 14 | 129 (42.3%) | 69 (22.6%) | 65 (21.3%) | 28 (9.2%) | 6 (2.0%) | 8 (2.6%) | 3.86 (1.26%) | 0.77 | 100.0 | 0.678 | 0.672 |

Note: Values for Items 1, 3, 4, 7, 9, 12 and 14 are reversed to calculate means and item difficulties. Corrected item-total correlations were calculated for model 1 (with all items and no correlations between items) and model 2 (without item 10 but no correlations between items).

items based on the evaluation of the original instrument [33]. CFA for this model showed standardized factor loadings between.305 (item 10) and.828 (item 9) with one item showing a factor loading below.40 (item 10; see Table 5). This indicated that item 10 did not fit to the predefined factor [58,59,67]. Values of RMSEA and TLI did not meet cut-offs [68,69] (see Table 6). We analyzed item 10 on the content level. In comparison to the other items, item 10 does not refer to people who have abortions, but to other people who make negative judgments about people having abortions. This item appears to measure attitudes towards abortion rather implicitly. Based on a discussion of these findings in conjunction with the item analysis for item 10, we decided to calculate an alternative model excluding item 10 (model 2). In model 2, factor loadings ranged from 0.437 (item 8) to 0.827 (item 9) (see Table 5). Model fits of model 2 were comparable to those of model 1, with RMSEA and TLI values remaining below the respective cut-off criteria. However, both AIC and BIC indicated a better fit for model 2 (see Table 6). Additionally, we calculated modification indices for all item pairs and found the highest index for the combination of item 8 and 13. The $\chi^2$ test statistic for freeing the covariance between those two items was 45.519 (degree of freedoms = 1). Upon further discussion within the study team, we identified a strong conceptual similarity between items 8

**Table 5. Factor loadings of two calculated models for the factor analysis of the German AAS (*n*=305 medical students).**

|        | Model 1 | Model 2 | Model 3 |
|--------|---------|---------|---------|
| Item 1 | 0.598 | 0.600 | 0.600 |
| Item 2 | 0.759 | 0.759 | 0.758 |
| Item 3 | 0.750 | 0.749 | 0.750 |
| Item 4 | 0.577 | 0.578 | 0.580 |
| Item 5 | 0.602 | 0.600 | 0.598 |
| Item 6 | 0.669 | 0.666 | 0.665 |
| Item 7 | 0.813 | 0.815 | 0.819 |
| Item 8 | 0.438 | 0.437 | 0.418 |
| Item 9 | 0.828 | 0.827 | 0.829 |
| Item 10 | 0.305 | – | – |
| Item 11 | 0.647 | 0.648 | 0.643 |
| Item 12 | 0.767 | 0.769 | 0.772 |
| Item 13 | 0.540 | 0.541 | 0.528 |
| Item 14 | 0.713 | 0.712 | 0.713 |

Notes: Model 1 is a one-factor model with no correlations between items. Model 2 is a one-factor model with no correlations between items and without item 10. Model 3 is a one-factor model with correlations between items 8 and 13 and without item 10. All models were calculated for the whole data set (*n*=305).

**Table 6. Fit indices of two calculated models for factor analysis of the German AAS (*n*=305 medical students).**

|                | Chi² [1] | df [2] | p value | RMSEA [3] | SRMR [4] | TLI [5] | CFI [6] | AIC [7] | BIC [8] | Ω [9] |
|----------------|----------|--------|---------|-----------|----------|---------|---------|---------|---------|-------|
| Recommendation |          |        | >0.05   | ≤0.06     | <0.08    | ≥0.90   | ≥0.90   | Model with the smallest value | | ≥0.70 |
| Model 1 | 252.1 | 77 | <0.001 | 0.086 | 0.051 | 0.895 | 0.911 | 11342 | 11498 | 0.910 |
| Model 2 | 235.9 | 65 | <0.001 | 0.093 | 0.053 | 0.894 | 0.912 | 10483 | 10628 | 0.913 |
| Model 3 | 187.4 | 64 | <0.001 | 0.079 | 0.045 | 0.923 | 0.937 | 10437 | 10585 | 0.904 |

Notes: Model 1 is a one-factor model with no correlations between items. Model 2 is a one-factor model with no correlations between items and without item 10. Model 3 is a one-factor model with correlations between items 8 and 13 and without item 10. All models were calculated for the whole data set (*n*=305). [1] Discrepancy chi-squared statistic, [2] Degrees of freedom, [3] Root mean square error of approximation, [4] Standardized Root Mean Squared Residual, [5] Tucker-Lewis Index, [6] Comparative Fit Index, [7] Akaike Information Criterion, [8] Bayesian Information Criterion, [9] McDonalds Omega.

and 13, as both items express essentially the same statement using different wording. Consequently, we calculated a third model (model 3) allowing the errors of item 8 and item 13 to correlate. Model 3 yielded factor loadings ranging from 0.418 to 0.819 (see Table 5) and demonstrated a better overall fit compared to models 1 and 2.

## Discussion

The original AAS is a brief instrument, which was developed and evaluated in 1983 in the United States for use by health educators helping students to explore their values concerning abortion [33]. The instrument was originally validated among high school and college students, but was later extended to a broader range of target populations [33]. Our study aimed to translate the English AAS into German, adapt it to Germanys' specific societal and legal context and evaluate its psychometric properties.

## Translation, adaptation and psychometric evaluation

The translation team quickly agreed on the translation of the AAS. No adaptations to the translated versions were necessary for items 2, 4, 7, 13, and 14. After minor adaptations, the introduction as well as items 1, 2, 4, 7, 8, 12, and 14 were well understood in cognitive interviews. For items 5, 6, and 10, one of two item versions was chosen, while items 3, 9, 11, and 13 were reworded based on participants' feedback. The final round of cognitive interviews confirmed that all items were comprehensible.

The German AAS demonstrated high acceptability. Most participants indicated having high AAS sum scores and were categorized as having a strong pro or moderate pro-abortion attitude (47.87% and 36.39% respectively). Only 3.5% were categorized as holding moderate or strong anti-abortion attitudes. Ceiling effects were observed, reflecting the sample's generally positive abortion attitude. Hollis and Morris found that, when asked about their abortion attitudes, most of their participants clustered at the low end or the high end even when applying a 7-point Likert scale to increase variance of responses [70]. However, response distribution in our data does not reflect the results of Swartz et al. (2020), who assessed the AAS in 504 nurses in California (United States) and found rather negative attitudes towards abortion [71]. They did not calculate sum scores but found that, for example, 52.4% ($n = 264$) strongly agreed with item 4 ("Abortion is wrong no matter what the circumstances are.") compared to 0.97% ($n = 2$) in our sample. In our sample, 82.3% ($n = 251$) strongly disagreed with this item. However, item analysis of the data of Swartz et al. discloses some contradictions [71]. For example, in their sample, 53.8% ($n = 271$) (strongly/somewhat) agreed that every conceived child has the right to be born (item 7) but 33.7% ($n = 397$) (strongly/somewhat) agreed that a pregnant woman who does not want a child should be encouraged to have an abortion (item 8). Alspaugh et al. (2022), who assessed abortion attitudes in women's health and neonatal nurses in the United States, found a more diverse picture [35]. In a sample of 1.820 nurses, 16% had strong pro-abortion attitudes, 32% had moderate pro-abortion attitudes, 29% were unsure, 11% had moderate anti-abortion attitudes and 13% had strong anti-abortion attitudes.

In our sample of German medical students, most preferred to work in the field of gynecology and obstetrics later (see table 2) and most have a pro-abortion attitude (see table 3). Based on the currently ongoing media-effective political and social discussions about abortion regulations in Germany, our sample might have been rather motivated to participate in our study, which is followed by a self-selection bias. In addition, due to the recruitment strategy applied, we could not reach all German medical students, and only $n = 305$ students participated in the survey. Consequently, our study population cannot be considered representative and no conclusions can be drawn about other groups. However, results of all described studies disclose that the AAS has the potential to differentiate between pro- and anti-abortion attitudes and is sensitive to the specific demography and background of different samples within a community.

Item characteristics for the German AAS were satisfying for most items, except for item 10 (original: "People should not look down on those who choose to have abortions"). This item showed a low corrected item-total correlations, an unfavourable response distribution (demonstrating a ceiling effect), weak inter-item correlation, and a factor loading below the recommended threshold of 0.40 in the CFA. These results suggest that item 10 does not measure the underlying concept [53,54,72]. In model 1, the a priori hypothesized one-factor structure was supported, although overall model fit indices were not fully satisfactory and item 10 showed the weakest factor loading. The study team concluded that the poor performance of item 10 likely reflects limited construct coverage rather than a mere statistical anomaly. After removing item 10 (model 2) and additionally allowing a correlation between item 8 and 13 based on modification indices and item's content (model 3), model 3 resulted in the best model fit with acceptable factor loadings [58,59,67]. It has to be stated that allowing correlated errors between Items 8 and 13 was a data-driven adjustment. All performed model adjustments represent post hoc modifications and, together with limitations due to the limited sample size and the potential sample bias, should therefore be interpreted tentatively and with caution. Freeing parameters and removing items improve model fit statistically but do not fully resolve the underlying measurement issues of the scale. Consequently, we consider the findings preliminary and recommend that the 13-itme version of the German AAS (without item

10) be subjected to further psychometric evaluation in larger and more diverse samples before any final modification of the instrument is adopted.

## Broader implications of the AAS

The World Health Organization considers abortion as a vital component of comprehensive sexual and reproductive health care [73]. However, physicians' attitudes toward abortion have been largely overlooked in empirical research. The successful translation and preliminary validation of the German version of the AAS highlights its potential as a culturally sensitive tool to evaluate attitudes towards abortions in sociocultural contexts where abortion remains ethically and politically sensitive. This adaptability is crucial for understanding how societal norms, laws, and individual beliefs intersect to shape attitudes towards abortions and abortion stigma in diverse populations. Our findings suggest that, while the overall factorial structure of the German AAS was supported, certain items (e.g., item 10 and the redundancy between item 8 and 13) require further psychometric refinement. These observations underline the importance of ensuring conceptual clarity and cultural appropriateness when applying the instrument in new populations. In the German context, the AAS may provide valuable insights into the attitudes of medical students and healthcare providers, helping to identify potential barriers to the provision of unbiased, person-centered abortion care. Understanding such attitudes early in medical training can inform the development of targeted educational interventions that address misconceptions and stigma [15,37,74]. For future applications, replication of the AAS in larger and more diverse samples is needed, along with longitudinal studies to assess the stability of abortion attitudes over time and their sensitivity to social and legal change. Such efforts will strengthen the evidence base for using the AAS as a reliable measure to be used by researchers, clinicians and policymakers to explore how community and professional attitudes toward abortion affect women's experiences, access to care, and decision-making. In this context, especially the assessment of experiences of stigma is relevant, particularly in contexts where restrictive laws and cultural norms contribute to feelings of shame, guilt, or fear among women seeking abortions.

## Strengths and limitations

A major strength of this study is that we provided the first tool to assess attitudes towards abortion in German for use across diverse settings and groups. We applied an established state-of-the-art translation procedure and used a qualitative approach to explore comprehensibility. Furthermore, we employed the German version of the AAS in a sample, which was large enough to perform robust psychometric analysis.

While the German AAS has shown promise, its limitations (such as the need to remove item 10) highlight the importance of ongoing validation and refinement. In our study, a self-selection bias among participants interested in the topic cannot be ruled out. Future studies should explore its application in more diverse and representative samples, including non-medical populations, healthcare providers, and patients, to ensure its broad utility and relevance. Furthermore, we encountered a high number of missing data within the demographic data of our participants, since those were assessed at the end of the survey. This limits the precise description of our sample. Since this was a secondary analysis of cross-sectional data, several psychometric parameters were not analyzable. For example, assessing convergent or divergent validity was not possible. Future validation studies could test convergent validity of the AAS by also applying the recently published 7-item Community Attitudes Abortion Scale (CAAS), which was developed from a survey of 1.533 women without a history of abortion seeking family planning services in the United States. It measures attitudes toward women who have abortions, anticipated behaviour towards a friend who has had an abortion, and attitudes towards abortion regulation [75].

## Conclusion

It is essential to measure attitudes towards abortion among medical students, physicians and other community groups using valid and reliable measures. We provide the first German measure for assessing this construct. The German AAS is a brief measure

that demonstrated high participant acceptance and acceptable psychometric properties. However, our findings indicate that certain items, particularly item 10 and the overlap between item 8 and item 13, require further refinement. Consequently, future validation should focus on the 13-item version of the scale, including examination of convergent and predictive validity in larger and more diverse samples. Despite these limitations, the German AAS shows potential as a tool for assessing attitudes towards abortion in Germany. This provides a foundation for identifying potential barriers in healthcare provision and informing the development of targeted educational and clinical interventions aimed at improving care for individuals facing unintended pregnancies.

## Supporting information

**S1 Appendix. Details on item translation and adaptation.**
(DOCX)

**S2 Appendix. Further demographic data and inter-item correlations.**
(DOCX)

**S3 Appendix. Checklist for reporting standards.**
(DOCX)

## Acknowledgments

We thank all participants of the online survey. We thank our cooperation partners (counselors in social support service and physicians providing abortions) for their critical and important feedback during the development of the online survey. We also thank Maja Lina Böcher for her help preparing this manuscript.

### Checklist for reporting statement

To report the results of this validation study, we used the Authors' Guidelines for Reporting Scale Development and Validation Results by Cabrera-Nguyen (see S3 Appendix). For assessment of comprehensibility as part of content validity, we used the COSMIN criteria (Consensus-based standards for the selection of health measurement instruments).

## Author contributions

**Conceptualization:** Anja Lindig.

**Data curation:** Anja Lindig, Mareike Rutenkröger, Mirja Baumgart.

**Formal analysis:** Anja Lindig, Eva Christalle.

**Funding acquisition:** Jördis Maria Zill.

**Investigation:** Anja Lindig, Mirja Baumgart.

**Methodology:** Anja Lindig, Mirja Baumgart.

**Project administration:** Anja Lindig.

**Supervision:** Eva Christalle.

**Writing – original draft:** Anja Lindig.

**Writing – review & editing:** Anja Lindig, Eva Christalle, Mareike Rutenkröger, Jördis Maria Zill, Mirja Baumgart.

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
