## [Decision Letter · Decision Letter 0]

30 Sep 2025

Dear Dr. Lindig,

Thank you for submitting your manuscript to PLOS ONE. After careful consideration, we feel that it has merit but does not fully meet PLOS ONE’s publication criteria as it currently stands. Therefore, we invite you to submit a revised version of the manuscript that addresses the points raised during the review process.

We look forward to receiving your revised manuscript.

Kind regards,

Marcia Leonardi Baldisserotto, Ph.D

Academic Editor

PLOS ONE

“This study was planned and conducted as part of the study “Person-centeredness in healthcare and support services for women with unwanted pregnancy” (CarePreg), which was funded by the German Federal Ministry of Health (Bundesministerium für Gesundheit, BMG) with the grant number 2520FSB113. Dr. Jördis Zill (Co-author) was principal investigator of the CarePreg Study and received the funding.”

“This study was planned and conducted as part of the study “Person-centeredness in healthcare and support services for women with unwanted pregnancy” (CarePreg), which was funded by the German Federal Ministry of Health (Bundesministerium für Gesundheit, BMG) with the grant number 2520FSB113.”

“This study was planned and conducted as part of the study “Person-centeredness in healthcare and support services for women with unwanted pregnancy” (CarePreg), which was funded by the German Federal Ministry of Health (Bundesministerium für Gesundheit, BMG) with the grant number 2520FSB113. Dr. Jördis Zill (Co-author) was principal investigator of the CarePreg Study and received the funding.”

4. In the online submission form you indicate that your data is not available for proprietary reasons and have provided a contact point for accessing this data. Please note that your current contact point is a co-author on this manuscript. According to our Data Policy, the contact point must not be an author on the manuscript and must be an institutional contact, ideally not an individual. Please revise your data statement to a non-author institutional point of contact, such as a data access or ethics committee, and send this to us via return email. Please also include contact information for the third party organization, and please include the full citation of where the data can be found.

Reviewers' comments:

Reviewer's Responses to Questions

**Comments to the Author**

1. Is the manuscript technically sound, and do the data support the conclusions?

Reviewer #1: Yes

Reviewer #2: Yes

2. Has the statistical analysis been performed appropriately and rigorously?

Reviewer #1: Yes

Reviewer #2: No

3. Have the authors made all data underlying the findings in their manuscript fully available?

Reviewer #1: Yes

Reviewer #2: Yes

4. Is the manuscript presented in an intelligible fashion and written in standard English?

Reviewer #1: Yes

Reviewer #2: No

Reviewer #1: The article has excellent practices on writing, argumentation and data analysis. I provide below some suggestions for enhancing the quality of the manuscript:

- Along all pages, consider the division of in long paragraphs in smaller ones, so that reading is more fluid;

- Review the use of the term "factorial structure" (it should be just "factor structure");

- When describing the methodology of congnitive interviews, consider providing examples of questions, so that this procedure may be better understood;

- In data analysis, describe how item difficulty was calculated;

- In the discussion, consider a a shorter descrption of the results observed in t he first paragraph, so that you can focus on implications for theory and practice in the subsequent text. The same recommendation applies to describing how you concluded for a third model of the instrument;

- In line 454, review use of commas (suggestion as follows): "it was, for example, not possible to" (...);

- The reference section has more than 70 references, which may not follow the journal's guidelines - consider review.

Reviewer #2: Line 211 reports an incorrect cut point for the KMO, indicating it as .05.

The study does not report the use of dimensionality testing procedures, such as parallel analysis or other recommended approaches.

Between lines 222 and 227, there is no mention of cut-off criteria for fit indexes or omega.

At line 298, the author refers to excluded cases as “data sets,” which is unusual. From this point onward, the language requires careful revision by a native speaker, as there are several colloquial expressions that compromise the formal tone of the manuscript.

The manuscript does not clearly describe the estimation process for item difficulty.

At line 342, a cut-off is mentioned, yet the methods section does not clarify which cut-off recommendations were followed. Subsequently, the author analyzes item pairs without making explicit the rationale for their selection. If modification indices guided these choices, this should be explicitly stated. Moreover, the writing from lines 345 to 354 lacks clarity and academic formality.

Table 3 presents three different models, one of which recommends the exclusion of an item. The authors should proceed with caution here, as they are suggesting modifications to an instrument based on a single sample of approximately 300 respondents. I strongly recommend revising the first discussion paragraph (lines 373–377), since it does not specify the target population for which the scale is intended. It should be restated that the measure is developed for medical students.

In the discussion (lines 407–408), it is not clear where the information on sample motivation is introduced. This should either be explicitly stated in the text or referenced in a table for ease of reader comprehension.

In lines 412–415 of the discussion, there is no clear statement regarding the factor loading of item 10, which appears to be missing.

Lines 418–422 would benefit from a clearer presentation of the scale’s limitations. Freeing parameters is a post hoc solution in CFA—it allows error to be modeled, but the underlying issue with the instrument persists. Therefore, the authors need to provide stronger recommendations regarding the interpretation of results, the functioning of the scale, and its shortcomings. The recommendation to remove an item (lines 420–422) is problematic, particularly given (1) the limited sample size, and (2) the potential sample bias, which the authors themselves later acknowledge. From lines 424–440, the manuscript discusses broader implications but fails to connect these to its own findings and to specify what is required for future applications of the AAS in the proposed context.

At lines 463–464, the manuscript refers to the psychometric investigation as “good” and “satisfying,” but these are subjective terms that do not adequately convey the findings. Furthermore, the discussion does not emphasize that this is only an initial step in adapting the scale, and that the results should be regarded as preliminary evidence of internal structure.

Finally, given the small sample size, the preliminary nature of the results, and other methodological limitations, the manuscript might be more appropriately suited for submission to a smaller, specialized journal.

**Do you want your identity to be public for this peer review?** For information about this choice, including consent withdrawal, please see our Privacy Policy

Reviewer #1: **Yes: ** Alessandro Antonio Scaduto

Reviewer #2: No

---

## [Author Response · Author response to Decision Letter 1]

28 Nov 2025

Thank you very much for the comments and suggestions, which had been send by the reviewers. We discussed every point carefully and answered each suggestion in the attached file "response to reviewers".

---

## [Editor Report · Decision Letter 1]

11 Dec 2025

Translation, adaptation and psychometric evaluation of the German version of the Abortion Attitude Scale – a secondary analysis of a cross-sectional study among medical students

PONE-D-25-12818R1

Dear, Anja Lindig

We’re pleased to inform you that your manuscript has been judged scientifically suitable for publication and will be formally accepted for publication once it meets all outstanding technical requirements.

Kind regards,

Marcia Leonardi Baldisserotto, Ph.D

Academic Editor

PLOS One
---

## [Editor Report · Acceptance letter]

PONE-D-25-12818R1

PLOS One

Dear Dr. Lindig,

I'm pleased to inform you that your manuscript has been deemed suitable for publication in PLOS One. Congratulations! Your manuscript is now being handed over to our production team.

Kind regards,

on behalf of

Dr. Marcia Leonardi Baldisserotto

Academic Editor

PLOS One